# Evaluating the Impact of Practical Training: A Study on Satisfaction and Drug Knowledge among Pharmacy Students

**DOI:** 10.3390/pharmacy12020069

**Published:** 2024-04-16

**Authors:** Nobuyuki Wakui, Shunsuke Shirozu, Yoshiaki Machida

**Affiliations:** Division of Applied Pharmaceutical Education and Research, Faculty of Pharmaceutical Sciences, Hoshi University, 2-4-41 Ebara, Shinagawa-ku, Tokyo 142-8501, Japan; s-shirozu@hoshi.ac.jp (S.S.); y-machida@hoshi.ac.jp (Y.M.)

**Keywords:** pharmacy students, practical training, satisfaction levels, drug knowledge acquisition, educational efficacy

## Abstract

Practical training stands as a crucial component in shaping pharmacy students, bridging the gap between classroom-based theoretical knowledge and its application in real-world contexts. This study investigates the correlation between the satisfaction levels experienced during practical training and the acquisition of knowledge, particularly focusing on drug names. Drawing from the National DataBase (NDB) open data provided by Japan’s Ministry of Health, Labour and Welfare, a survey centered on the top 100 dispensed drugs was crafted. A correlation analysis was conducted between the satisfaction scores gathered from students and their depth of knowledge on drug names. Intriguingly, despite observing a significantly high satisfaction level during the practical training, there was no marked correlation between this satisfaction and the proficiency in recalling drug names after the training. Furthermore, the volume of daily prescriptions handled did not significantly impact this knowledge. The findings underscore the notion that high satisfaction during training does not necessarily guarantee a thorough understanding of the subject matter. This sheds light on the importance of not solely relying on satisfaction metrics in training programs and ensuring a holistic, in-depth educational approach.

## 1. Introduction

Pharmacy students in Japan undergo 11 weeks of pharmacy practice training in their fifth year [1]. Prior studies have shown variations in both the skills acquired and the attitudes towards practical training among students, as well as among the pharmacies themselves [2], despite the standardized curriculum.

This mandatory training is executed across Japan for all fifth-year pharmacy students. Although there have been several evaluations of the skills and attitudes of pharmacy students [3,4,5,6,7,8], the relationship between the satisfaction levels with the training and the subsequent knowledge acquisition remains underexplored.

The pharmacies in Japan have a wide range of medical drugs at their disposal, given their ability to accept prescriptions from all clinics and hospitals [9]. Yet, about 70% of them predominantly cater to patients from nearby medical institutions [10]. As such, students trained in a pharmacy near a clinic specializing in, say, diabetes will predominantly encounter prescriptions related to that ailment. Furthermore, the type of medical institution, categorized by the number of beds, affects the prescriptions a pharmacy receives [11,12]. Given that pharmacy students are randomly assigned to a dispensing pharmacy based on their residential area, their exposure to different types of drugs and medical institutions varies. This could influence both their satisfaction with the training experience and the knowledge they gain.

Studies from other disciplines have explored the relationship between satisfaction levels and knowledge acquisition. For example, research on dental students’ e-learning satisfaction revealed that self-efficacy and learning styles significantly influence educational satisfaction, potentially affecting learning outcomes [13]. Similarly, research on the public sector’s synchronous on-the-job online training examined the correlation between learning motivation and satisfaction, underscoring the need to align educational practices with learners’ needs to enhance learning effectiveness [14]. These insights from various educational domains underline the complex nature of satisfaction and knowledge acquisition, suggesting the value of further investigation in pharmacy education.

Generally, it is assumed that if pharmacy students report high satisfaction following their practical training, the training site was likely effective. However, since practical training is primarily a learning opportunity, it is crucial to examine whether this satisfaction correlates with actual knowledge gains. This study proposes the need to assess not only satisfaction levels but also the accompanying knowledge acquisition to fully evaluate the training’s effectiveness.

This study was designed to investigate the link between pharmacy students’ satisfaction with their practical training and the knowledge they acquire, a facet that has received little attention in prior research. By assessing the relationship between satisfaction levels and knowledge acquisition, we aimed to shed light on the learning status of pharmacy students and sought to gain new insights.

## 2. Materials and Methods

### 2.1. Study Design

This longitudinal study involved Hoshi University pharmacy students who were scheduled to undergo pharmacy practice training in 2021. Before starting their practical training and after its completion, the students were asked to complete a questionnaire using Google Forms. The primary objective was to assess the impact of the practical training on the students’ knowledge of drug names. By analyzing the results of the questionnaires before and after the training, we could determine the percentage change in the students’ knowledge scores related to drug names. Additionally, we sought to gauge the satisfaction levels of the students with their training experience. The entire duration of the survey spanned from 21 February 2021 to 17 May of the same year. Considering the students’ time, the questionnaire was designed to be completed in approximately 10 min, a duration confirmed by pre-testing. Then, following their on-the-job experience at the pharmacy, a post-training questionnaire was administered from 10 to 17 May 2021. This was designed to evaluate both the improvement in their drug name knowledge and their overall satisfaction with the training program.

To facilitate participation, eligible students received an email outlining the survey’s purpose and procedure. This study emphasized voluntary participation, with respondents assenting to partake at the beginning of the survey. Google Forms was chosen as the platform, and students accessed the questionnaire via a URL link dispatched by the research team.

### 2.2. Practical Training Framework for Pharmacy Students in Japan

The practical training is a comprehensive experience designed not only to deepen their knowledge of drugs but also to enhance their communication and problem-solving abilities as pharmacists. Through their interactions with various individuals, students learn to apply their academic understanding in real-world scenarios, gaining insights into the daily roles and challenges of a pharmacist [15]. The program commences with 11 weeks of training in a pharmacy, where students are introduced to various aspects, including prescription-based drug treatments, home medical care, and self-medication practices.

### 2.3. Questionnaire/Survey of Knowledge of Drug Names

The first survey was administered before the start of the pharmacy training, gathering demographic information such as age and sex along with a test on drug name knowledge. In the drug name test, students were presented with the names of drugs and required to identify their effects. A student’s response was counted if they could correctly recognize both the drug’s name and its effects. Following the pharmacy training, a second survey was issued. This survey revisited the drug name test from the initial survey and further explored the number of prescriptions students worked on during their training and their overall satisfaction with the practice experience. To do this, we designed a survey encompassing questions on drug name knowledge, based on the top 100 drugs dispensed in Japan as listed by the Ministry of Health, Labour and Welfare [11,12]. For the questionnaire, drug names were derived from the top 100 best-selling oral medications in Japan as documented by the National DataBase (NDB) open data [16]. From this top 100 list, we randomly selected 50 drugs to be featured in the survey. A detailed overview of the drugs included in the survey can be accessed in Appendix A.

### 2.4. Satisfaction Survey on Pharmacy Practice Training

The survey was conducted using a 5-point Likert scale. Participants were asked to select their feedback regarding the practical training from the following options: “Extremely Satisfied”, “Satisfied”, “Neutral”, “Dissatisfied”, and “Extremely Dissatisfied”. These responses were scored as follows to calculate satisfaction scores: Extremely Satisfied was scored as 5 points, Satisfied as 4 points, Neither as 3 points, Dissatisfied as 2 points, and Extremely Dissatisfied as 1 point.

### 2.5. Statistical Analysis

The statistical software SAS (version 9.4, SAS Institute, Inc., Cary, NC, USA) was used for data analysis. The total drug score was calculated by converting the survey results of the knowledge of drug name test to a full score of 100 points with 2 points for each question, and the mean ± standard deviation was calculated. A paired *t*-test was performed to compare the differences in mean values before and after the training. Spearman’s rank correlation coefficient was used to calculate the correlations separately: one between the change in drug knowledge scores and the five levels of training satisfaction, and another between the change in drug knowledge scores and the number of prescriptions. The level of significance in all tests was *p* < 0.05. The number of missing values was 0 because we prepared the Google forms in such a way that the survey would not end if any section was left unanswered.

### 2.6. Ethical Considerations

This study was conducted with the approval of the ethics committee of the Hoshi Pharmaceutical University (approval number: 2020-12). On the first page of the survey form, explanations about answering the form, voluntary participation, anonymity, and confidentiality were given and informed consent was obtained online from the study participants.

## 3. Results

### 3.1. Participant Information

Of the 105 individuals invited to participate in the study, 78 responded, giving a participation rate of 74.3%. Of these respondents, 83.5% were female. The 74.3% participation rate reflects individuals who agreed to participate before the practical training and were also sent the post-training questionnaire. These participants completed both the pre-test and the post-test. The overall average satisfaction score with practical training was 4.45, with men scoring 4.46 and women 4.45, indicating no significant difference in satisfaction levels between genders (*p* = 0.97). The average age of the participants was 22.19 ± 0.38 years, with males averaging 22.23 years and females 22.18 years, showing no significant difference between genders (*p* = 0.85).

### 3.2. Changes in Scores before and after the Practical Pharmacy Training

After the training, participants showed a notable increase in their drug knowledge scores compared to their scores before the training (*p* < 0.001). This improvement in drug name knowledge was evident for everyone and was similarly significant when analyzed by gender (Table 1).

### 3.3. Relationship between Satisfaction with Practical Training and Drug Knowledge Scores

The results of the questionnaire after the pharmacy training are shown in Table 2. The mean level of satisfaction with the practical training was 4.5 ± 0.7. The correlation between the change in drug knowledge scores from before to after the practical training and student satisfaction was not significant (r = −0.03, *p* = 0.81). The highest average number of prescriptions handled daily in the practical training group was 10–19, followed by 20–29 and 30–39. Similarly, no significant difference was observed in the relationship between the average number of prescriptions handled daily and the change in drug knowledge scores from before to after the practical training (r = 0.03, *p* = 0.81).

Table 2 shows the relationship between satisfaction with practical training and drug knowledge scores. The prompt for satisfaction was ‘Please answer the satisfaction level of the practical training on a 5-point scale’. For number of prescriptions, the prompt was ‘Please answer the number of prescriptions per day at the pharmacy where the implementation was carried out’. Practical satisfaction and number of prescriptions per day were not associated with increased drug scores.

## 4. Discussion

In this study, we evaluated the relationship between satisfaction with practical training and the knowledge about drug names acquired by pharmacy students during a clinical placement in a pharmacy. The mean level of satisfaction with the practical training was very high. However, there was also no significant difference in the relationship between practice training satisfaction and drug knowledge score after the practical training. In addition, there was also no significant difference in the relationship between the average number of prescriptions handled daily and drug knowledge score after the practical training.

Our study indicates that high levels of satisfaction with practical training do not necessarily correlate with an increase in drug knowledge among pharmacy students. This finding is consistent with broader educational trends where the quality of the educational environment and interactions, rather than the mere acquisition of knowledge, play a crucial role in shaping students’ experiences and satisfaction [7]. While the introduction of the 6-year pharmaceutical education system in our country has been associated with improved evaluations of pharmacy practice programs, our results suggest that satisfaction is influenced by factors beyond direct knowledge gain, such as the support system at the training site and the quality of dialogue with patients. This underlines the importance of a comprehensive approach in pharmacy education that integrates training sites, universities, and communities to cultivate pharmacists who can make significant contributions to the field. Additionally, it implies that high satisfaction levels should not be misconstrued as indicative of knowledge acquisition, suggesting the need for a nuanced understanding of how satisfaction and learning outcomes are interconnected in pharmacy education.

In the context of similar studies, high satisfaction in educational or clinical settings has not consistently translated into enhanced knowledge or competence. Research involving practicing pharmacists, for instance, has revealed discrepancies between their self-assessed knowledge and actual knowledge levels, suggesting that perceived competence might not fully reflect true expertise [17]. While this particular study focuses on pharmacists, a similar pattern is observed in our research with pharmacy students, indicating that high satisfaction with practical training does not necessarily lead to increased drug knowledge.

In our endeavor to elucidate the interplay between practical training satisfaction and the acquired knowledge of drug names among pharmacy students, some intriguing patterns emerged. One of the salient findings is the high average satisfaction rating regarding the practical training, implying that the majority of students found the experience beneficial and fulfilling. Nonetheless, it is worth noting that this heightened sense of satisfaction did not necessarily translate to a corresponding increase in drug name knowledge. This suggests that while the students might appreciate the hands-on exposure and learning environment of the training, this appreciation does not guarantee a proportionate expansion in their drug name lexicon.

Our study results indicate that there is no significant correlation between the volume of prescriptions handled per day during training and the change in drug knowledge scores post training. This suggests that merely increasing the number of prescriptions students interact with may not improve their drug knowledge. Therefore, it appears that rather than the quantity of prescriptions, a thorough understanding of and engagement with each prescription are more crucial.

The finding that there is no correlation between satisfaction and knowledge scores is particularly intriguing, suggesting that regardless of the training environment, students tend to improve their knowledge of drug names throughout pharmacy practice. This observation prompts further reflection on the nature of learning and satisfaction in pharma-cy education. Despite the high levels of reported satisfaction, this does not necessarily equate to increased knowledge of drugs, indicating that satisfaction may be influenced by factors other than knowledge acquisition. For instance, interpersonal dynamics, such as the kindness of or compatibility with the supervising pharmacist, could contribute to higher satisfaction levels. Furthermore, high satisfaction levels during pharmacy practice might not only reflect the nature of the immediate training experience but also influence students’ decisions to pursue a career in pharmacy. This underscores the significance of the training experience, as it can have a lasting impact on students’ future professional choices.

The purpose of this study was to suggest the relationship between satisfaction and knowledge improvement, and, therefore, a satisfaction survey was conducted using a 5-level Likert scale to obtain quantitative data to evaluate the presence of this relationship. In future research, it will be necessary to conduct surveys considering factors that influence knowledge improvement and satisfaction, referring to similar overseas studies conducted in the past [18,19].

Further research should investigate factors such as the diversity of drugs in each prescription, the extent of patient interaction during medication counselling, and the characteristics of pharmacy practice settings that facilitate effective learning. A deeper understanding of these factors could lead to improvements in training methods that better enhance students’ pharmaceutical knowledge. In light of these findings, it is vital for training programs to emphasize not just the sheer number of drug names students are exposed to, but also a deeper understanding of those drugs as a whole. High satisfaction ratings in training are commendable, but it should be ensured that students can grasp a broader spectrum of drug-related knowledge.

Our study, while insightful, had a primary limitation: it exclusively focused on the knowledge of drug names. This narrow lens potentially overlooks broader aspects of pharmaceutical understanding, such as drug mechanisms, side effects, or therapeutic uses. While our results shed light on drug name recall, other areas of drug knowledge might be influenced differently by the training. Future research should adopt a more comprehensive approach, encompassing various facets of pharmaceutical knowledge to provide a holistic view of practical training’s impact on pharmacy students.

This discussion offers an in-depth analysis of the findings and proposes potential areas of improvement in training programs for future consideration.

## 5. Conclusions

High satisfaction rates during practical training might indicate a positive student experience. Yet, our findings underline that high satisfaction does not necessarily correspond to deep knowledge acquisition. It is crucial not to conflate contentment with comprehensive understanding. Training programs should aim to strike a balance between fostering student satisfaction and ensuring rigorous educational outcomes.

## Figures and Tables

**Table 1 pharmacy-12-00069-t001:** Changes in the scores before and after practical training.

Practical Training (*n* = 78)		Before Score	After Score	Change	*p*-Value
Total		70.7 ± 25.5	84.8 ± 9.9	14.1 ± 22.4	*p* < 0.001
Sex	Male (*n* = 13)	67.7 ± 21.8	81.8 ± 15.7	14.2 ± 11.1	*p* < 0.001
	Female (*n* = 65)	71.3 ± 26.3	85.4 ± 8.4	14.1 ± 24.0	*p* < 0.001

**Table 2 pharmacy-12-00069-t002:** Distribution of training satisfaction and number of prescriptions.

	Category	*n* (%)	Correlation	*p*-Value
Satisfaction			r = −0.03	0.81
	Extremely Satisfied	43 (55.1%)		
	Satisfied	27 (34.6%)		
	Neither	8 (10.3%)		
	Dissatisfied	0 (0%)		
	Extremely Dissatisfied	0 (0%)		
Number of prescriptions			r = 0.03	0.81
	1–9	7 (9.0%)		
	10–19	24 (30.8%)		
	20–29	23 (29.5%)		
	30–39	15 (19.2%)		
	40–49	6 (7.7%)		
	≥50	3 (3.9%)		

## Data Availability

The data presented in this study are available on request from the corresponding author.

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
