# Peer review of "Evaluating the Impact of Practical Training: A Study on Satisfaction and Drug Knowledge among Pharmacy Students"

_pharmacy, 2024, doi:10.3390/pharmacy12020069_

Round 1

Reviewer 1 Report

Comments and Suggestions for Authors

The authors attempted to evaluate the impact of practical training by investigating the correlation between the satisfaction of Japanese pharmacy students with their practical training and their drug name knowledge scores. However, I have several concerns that I’d like to address in this the following concerns.

<Major comments>

1. Introduction

It is unclear what the gap on line 46 refers to.

Does it refer to:

a) The gap between the conditions of the pharmacies where training takes place (proximity to diabetes clinics, size of medical institutions, number of prescriptions, etc. ) and the knowledge acquired by students?

b) The gap between pharmacy training content and the knowledge acquired by students?

c) The gap between students' subjective satisfaction and the knowledge they acquire?

I guess that the authors focused on "how knowledge acquisition varies depending on student satisfaction."

If my understanding is correct, the authors should clarify this study’s purpose, if not, the authors may need to provide additional data (pharmacy background, contents of training, etc.) and revise the entire paper accordingly.

2. Reference to the Hospital in the article

This study focused on training at the community pharmacy. However, the mentions of hospital training in lines 26 and 74-78 raise questions about their relevance to this investigation. If these references are pertinent, they should be revised to explicitly connect them to this study. Conversely, if they are extraneous, their removal should be considered.

3. Methods (knowledge of drug name)

How was the drug name test conducted? Did students have to respond by stating the effects of the drugs after seeing their names? Or were they required to indicate whether they were familiar with the drug names? I recommend that the authors provide more details as this will affect the discussion.

4. Results

To understand how much knowledge of drug names students acquired during their training period at the pharmacy, I believe it's necessary for the authors to also investigate the relationship between the difference in scores before and after training and student satisfaction. What are the authors' thoughts on this?

5. Discussion/Conclusion

Lines 169 onwards contain statements that are not based on the results. I suggest that the research group engage in further discussion regarding the interpretation of the results. Subsequently, please provide appropriate discussion aligned with the study's objectives, accompanied by relevant literature citations.

<Lines169-188>

This study did not provide information regarding the characteristics of the pharmacies where students trained or the relationship between these characteristics and the number of prescriptions. Therefore, it cannot stated that "the sheer number of prescriptions might not provide diverse exposure to drug names if many prescriptions are repetitive or concern a limited set of drugs." Additionally, this study did not investigate what kind of training students received at the pharmacies and what aspects they prioritized. Therefore, "However, our results suggest that most students might be taking the quicker, less thoughtful approach." is unfounded.

The finding that there is no correlation between satisfaction and knowledge scores may be an aspect that readers find intriguing. The authors should focus on this primary result and provide further discussion. Considering a) the significant increase in knowledge after training, and b) the lack of correlation between satisfaction, prescription volume, and knowledge score in this study, it appears that regardless of the training environment, students seem to improve their knowledge of drug names throughout pharmacy practice.

<Lines189-203>

Based on the above, please reconsider the contents provided.

<Minor comments>

1. introduction (lines47-4950-52)

“To do this ….Welfare[11,12]” should be moved to the methods section.

2. Methods (Statistical analysis)

In Table 2, the correlation between the number of prescriptions and drug scores is presented, so this should also be added to the Methods section.

3. Line 141 by 20-23? à by 20-29

4. Line 144 Table 3. ? à Table 2.

Author Response

Dear Reviewer 1,

Thank you for reviewing my manuscript. I appreciate your feedback, and I have made the necessary revisions based on your suggestions. Please find the attached response document for your review.

Best Regard,

Nobuyuki Wakui

Reviewer 2 Report

Comments and Suggestions for Authors

The paper describes satisfaction of pharmacy training and the effect on drug knowledge. The paper is well written and can be interesting to people working with pharmacy education. I have the following comments:

Line 26: here it is mentioned that the students undergo 11 weeks of hospital training. The focus of the paper is however on the pharmacy training. I suggest that you add the information in the introduction that the students complete both pharmacy training and hospital training during their education.

Line 34. Although the correlation between satisfactory levels and knowledge acquisition perhaps is scarcely studied in the pharmacy context, there are reports in literature about this in other disciplines. The introduction would be strengthened by adding what is found in literature regarding this correlation, although in other subject areas. Generally, satisfaction among learners is considered to have a major impact on learning achievements and academic success. Perhaps other satisfactions than satisfaction with pharmacy training could affect drug product knowledge. The introduction must more clearly describe the hypothesis and the rationale for investigating the correlation between pharmacy training satisfaction and drug knowledge. What is the theoretical framework?

Line 63: Please be consistent when writing dates. February 21, 2021 to May 17 of the same year.

Line 69: Please clarify what ”their” knowledge refers to, i.e. the students’ knowledge.

Line 64: was the survey tested before distributed?

Line 97: demographic information was collected, was there more information than age and sex? Why is not age presented in the results? Where there any difference in score or satisfaction in  different age groups?

Line 136: did everybody answer both the surveys? Does the response rate of 74.3% include answering both surveys? If there were students who only answered the first survey, were they invited to the second one? Please clarify this in the paper.

Line 137: ”the practical training groups” - does this mean after the pharmacy training?

Please revise the numbering of the tables. Table 3 should be table 2.

Where three any difference in satisfaction between men and women? Between different age groups?

Where the score before and after practical training considers good or bad? What was expected regarding drug product knowledge?

The discussion lacks reference to other studies. This has to be added to put the results of the study in a broader context. 

Author Response

Dear Reviewer 2

Thank you for reviewing my manuscript. I appreciate your feedback, and I have made the necessary revisions based on your suggestions. Please find the attached response document for your review.

Best Regard,

Nobuyuki Wakui

Reviewer 3 Report

Comments and Suggestions for Authors

The manuscript aims to presents the results of the study of student satisfaction and drug knowledge in the group of pharmacy students attending mandatory practical training in hospital pharmacy. The manuscript shows the relation between the increased knowledge and contributing factors, i.e. student satisfaction and number of prescription handled by students. The brief report showed that following the practical training students knowledge increased and that this was independent of students satisfaction. The strength of the manuscript are large sample of students, clearly indicated aims and research gap, neat data presentation, appropriate statistical analysis. Most literature is recent and on the topic.

However, there are some issues that decrease the quality of the manuscript:

1)     The methodology section does not include description of satisfaction measurements

2)     Section 2.3 repeats what was stated on section 2.1

3)     In section 3.1 it is not stated whether all participants that filled the pretest also completed the post-test

4)     In section 2.1 authors site 3 articles (13-15) in order to back up the use of Google Forms as a too for running a survey and this is unnecessary. These references could be omitted

5)     Most importantly, the discussion fails to discuss the presented results with other recent studies or to explain why and how the increase in drug knowledge occurred

6)     I believe that the method for measuring the satisfaction in 5-point scale has not been properly chosen. Instead more factors should be included in order elucidate which elements really contribute to the observed educational improvement of students’ knowledge of drug names. Students could be satisfied for many reasons: education in real-life settings, good working hours, or other practical things, that only vaguely influence learning. However what could really be the reason for the observed improved knowledge could be e.g. level of supervision, feedback from trainer, tools available to students at workplace, variety of experiences encountered during the training, etc. Role of these factors, especially feedback, has been well documented, by e.g. John Hattie Visible feedback books. A proper tool that would include these factors should be employed. Authors could e.g. use the method used here: DOI:10.1177/0897190013489575. Authors could also use some mixed methods, i.e. include interviews with selected students to elucidate how did the knowledge increase from the perspective of the learner, not just the teacher.

I think that the manuscript should be revised in order to be more meaningful and better discuss the results with previously published findings.

Author Response

Dear Reviewer 3

Thank you for reviewing my manuscript. I appreciate your feedback, and I have made the necessary revisions based on your suggestions. Please find the attached response document for your review.

Best Regard,

Nobuyuki Wakui

Round 2

Reviewer 1 Report

Comments and Suggestions for Authors

Thank you for the revisions. I think adding additional information and descriptions would be useful for readers.

<Minor comments>

1. Line 257 "pharma-cy" → "pharmacy"

I recommend that you carefully review the entire article for such errors.

Reviewer 2 Report

Comments and Suggestions for Authors

Thank you for your revised manuscript. I think the authors have addressed my comments satisfactorily.

Reviewer 3 Report

Comments and Suggestions for Authors

The authors addressed all my comments. I reccomend accepting the manuscript in present form and have no further comments or suggestions.